# APBench: A Unified Benchmark for Availability Poisoning Attacks and Defenses

**Tianrui Qin**[*a,b,d], **Xitong Gao**[*†a,c], **Juanjuan Zhao**[a], **Kejiang Ye**[a,c], **Cheng-Zhong Xu**[e]

[a] Shenzhen Institutes of Advanced Technology, Chinese Academy of Sciences, Shenzhen, China.
[b] University of Chinese Academy of Sciences, Beijing, China.
[c] Shenzhen University of Advanced Technology, Shenzhen, China.
[d] Tencent Security Platform, Tencent, Shenzhen, China.
[e] University of Macau, Macau S.A.R., China.

**Reviewed on OpenReview:** `https://openreview.net/forum?id=igJ2XPNYbJ`

## Abstract

The efficacy of availability poisoning, a method of poisoning data by injecting imperceptible perturbations to prevent its use in model training, has been a hot subject of investigation. Previous research suggested that it was difficult to effectively counteract such poisoning attacks. However, the introduction of various defense methods has challenged this notion. Due to the rapid progress in this field, the performance of different novel methods cannot be accurately validated due to variations in experimental setups. To further evaluate the attack and defense capabilities of these poisoning methods, we have developed a benchmark — APBench for assessing the efficacy of adversarial poisoning. APBench consists of 11 state-of-the-art availability poisoning attacks, 8 defense algorithms, and 4 conventional data augmentation techniques. We also have set up experiments with varying different poisoning ratios, and evaluated the attacks on multiple datasets and their transferability across model architectures. We further conducted a comprehensive evaluation of 2 additional attacks specifically targeting unsupervised models. Our results reveal the glaring inadequacy of existing attacks in safeguarding individual privacy. APBench is open source and available to the deep learning community[1].

## 1 Introduction

Recent advancements of deep neural networks (DNNs) (LeCun et al., 2015; Schmidhuber, 2015; He et al., 2016) heavily rely on the abundant availability of data resources (Deng et al., 2009; Russakovsky et al., 2015; Karras et al., 2020). However, the unauthorized collection of large-scale data through web scraping for model training has raised concerns regarding data security and privacy. In response, a new paradigm of practical and effective data protection methods has emerged, known as availability poisoning attacks (APA) (Tao et al., 2021; Yuan & Wu, 2021; Fowl et al., 2021; Huang et al., 2021; Wu et al., 2023; Fu et al., 2022; Ren et al., 2023; He et al., 2023; Sandoval-Segura et al., 2022b; Feng et al., 2019; Yu et al., 2022; He et al., 2023; Ren et al., 2023) or unlearnable example attacks. These poisoning methods inject small perturbations into images that are typically imperceptible to humans, in order to hinder the model's ability to learn the original features of the images. Recently, the field of deep learning has witnessed advancements in defense strategies (Liu et al., 2023; Qin et al., 2023; Dolatabadi et al., 2023; Huang et al., 2021) that hold the potential to challenge APAs, thereby undermining their claimed effectiveness and robustness. These defenses reveal the glaring inadequacy of existing APAs in safeguarding individual privacy in images. Consequently, we anticipate an impending arms race between attack and defense strategies in the near future.

However, evaluating the performance of these new methods across diverse model architectures and datasets poses a significant challenge due to variations in experimental settings of recent literatures. In addition, researchers face the

---

[*]Equal contribution.
[†]Correspondence to Xitong Gao (`xt.gao@siat.ac.cn`).
[1]`https://github.com/lafeat/apbench`.

daunting task of staying abreast of the latest methods and assessing the effectiveness of various competing attack-defense combinations. This could greatly hamper the development and empirical exploration of novel attack and defense strategies.

To tackle these challenges, we propose APBench, a benchmark specifically designed for availability poisoning attacks and defenses. It involves implementing poisoning attack and defense mechanisms under standardized perturbations and training hyperparameters, in order to ensure fair and reproducible comparative evaluations. APBench comprises a range of availability poisoning attacks and defense algorithms, and commonly-used data augmentation policies. This comprehensive suite allows us to evaluate the effectiveness of the poisoning attacks and the robustness of the defense strategies across multiple datasets and model architectures.

Our contributions can be summarized as follows:

- An open source benchmark for state-of-the-art availability poisoning attacks and defenses, including 9 supervised and 2 unsupervised poisoning attack methods, 8 defense strategies and 4 common data augmentation methods.

- We conduct a comprehensive evaluation competing pairs of poisoning attacks and defenses under standardized experimental settings.

- We carried out experiments across 4 publicly available datasets, and also extensively examined scenarios of partial poisoning, increased perturbations, the transferability of attacks to 4 CNN and 4ViT models under various defenses, and unsupervised learning. We provide visual evaluation tools such as t-SNE, Shapley value map and Grad-CAM to qualitatively analyze the impact of poisoning attacks.

The aim of APBench is to serve as a catalyst for facilitating and promoting future advancements in both availability poisoning attack and defense methods. By providing a platform for evaluation and comparison, we aspire to pave the way for the development of future availability poisoning attacks that can effectively preserve utility and protect privacy.

## 2 Related Work

### 2.1 Availability Poisoning Attacks

Availability poisoning attacks (APAs) belong to a category of data poisoning attacks (Goldblum et al., 2022) that adds a small perturbation to images, that is often imperceptible to humans. The objective contrasts with the malicious intent of traditional data poisoning – instead, the purpose of these perturbations is to **protect individual privacy** from deep learning algorithms, preventing DNNs from effectively learning the features present in the images. The attacker's goal is to render their data unlearnable with perturbations, hindering the unauthorized trainer from utilizing the data to learn models that can generalize effectively to the original data distribution. **The intent of APAs is thus benign rather than malicious** as generally assumed of data poisoning attacks. We typically assume that the attacker publishes (a subset of) the images, which get curated and accurately labeled by the defender to train on, typically without consent from the attacker.

Formally, consider a source dataset comprising original examples $\mathcal{D}_{\text{clean}} = \{(\mathbf{x}_1, y_1), \ldots, (\mathbf{x}_n, y_n)\}$ where $\mathbf{x}_i \in \mathcal{X}$ denotes an input image and $y_i \in \mathcal{Y}$ represents its label. The objective of the attacker is thus to construct a set of availability perturbations $\boldsymbol{\delta}$, such that models trained on the set of *availability poisoned examples* $\mathcal{D}_{\text{poi}}(\boldsymbol{\delta}) = \{(\mathbf{x} + \boldsymbol{\delta}_{\mathbf{x}}, y) \mid (\mathbf{x}, y) \in \mathcal{D}_{\text{clean}}\}$ are expected to perform poorly when evaluated on a test set $\mathcal{D}_{\text{test}}$ sampled from the distribution $\mathcal{S}$:

$$\max_{\boldsymbol{\delta}} \mathbb{E}_{(\mathbf{x}_i, y_i) \sim \mathcal{D}_{\text{test}}}[\mathcal{L}(f_{\boldsymbol{\theta}^\star(\boldsymbol{\delta})}(\mathbf{x}_i), y_i)],$$
$$\text{s.t. } \boldsymbol{\theta}^\star(\boldsymbol{\delta}) = \operatorname*{argmin}_{\boldsymbol{\theta}} \mathbb{E}_{(\hat{\mathbf{x}}_i, y_i) \sim \mathcal{D}_{\text{poi}}(\boldsymbol{\delta})} \mathcal{L}(f_{\boldsymbol{\theta}}(\hat{\mathbf{x}}_i), y_i), \tag{1}$$

where $\mathcal{L}$ denotes the loss function, usually the softmax cross-entropy loss. In order to limit the impact on the original utility of images, the perturbation $\boldsymbol{\delta}_i$ is generally constrained within a small $\epsilon$-ball of $\ell_p$ distance.

In the realm of adversarial machine learning, a key challenge is to enforce a small perturbation budget while maximizing the disruptive impact on the target model. To address this challenge, recent methods have typically constrained their

perturbations within $\ell_p$ perturbation, where $p \in \{0, 2, \infty\}$. This constraint ensures that the perturbations are relatively imperceptible to humans but can still have a significant impact on the behavior of the target model.

First, we introduce the APAs examined in APBench below:

- **DeepConfuse (DC)** (Feng et al., 2019): DC is a precursory APA which uses autoencoders to generate unlearning perturbations. Its key idea involves employing an autoencoder-like network to capture the training trajectory of the target model and adversarially perturbing the training data.

- **Neural tangent generalization attacks (NTGA)** (Yuan & Wu, 2021): NTGA approximates the targeted model training as a Gaussian process (Jacot et al., 2018), and leverages this surrogate to find better local optima with improved transferability.

- **Error-minimizing attack (EM)** (Huang et al., 2021): EM trains a surrogate model by minimizing the error of images relative to their original labels, generating perturbations that minimize the errors and thus render the perturbed images unlearnable. The authors of EM introduce the threat model of availability poisoning attacks, highlighting their role as a mechanism for privacy protection.

- **Robust error-minimizing attacks (REM)** (Fu et al., 2022): REM is a natural extension of EM to generate attack perturbations with adversarial training of the surrogate model. This approach aims to enhance attack robustness against adversarial training-based defenses.

- **Hypocritical (HYPO)** (Tao et al., 2021): HYPO, similar to EM, generates error-minimizing perturbations akin to those used in EM but employs a pretrained surrogate model.

- **Targeted adversarial poisoning (TAP)** (Fowl et al., 2021): TAP achieves availability poisoning by generating targeted adversarial examples of non-ground-truth labels of pre-trained models. It shows that adversarial attacks can also be effective APAs.

- **Linear-separable poisoning (LSP)** (Yu et al., 2022): LSP is a prescription-based APA which algorithmically generates randomly initialized linearly separable color block perturbations, enabling effective APAs without requiring surrogate models or excessive computational overhead.

- **Autoregressive Poisoning (AR)** (Sandoval-Segura et al., 2022b): AR, similar to LSP, is prescription-based, and does not require additional surrogate models. It fills the initial rows and columns of each channel with Gaussian noise and uses an autoregressive process to fill the remaining pixels, generating random noise perturbations that are linearly separable.

- **One-Pixel-Shortcut (OPS)** (Wu et al., 2023): OPS is a targeted availability poisoning attack that perturbs only one pixel per image, generating an effective availability poisoning attack against traditional $\ell_{\{2,\infty\}}$-bounded adversarial training methods.

- **Contrastive Poisoning (UCL)** (He et al., 2023) uses the contrastive loss (*e.g.*, the InfoNCE loss) to generate perturbations that are effective under unsupervised learning settings.

- **Transferable Unlearnable Examples (TUE)** (Ren et al., 2023) boosts the transferability of unlearnable examples with linear separable poisons produced with the class-wise separability discriminant.

These various attack strategies underscore the complexity and multi-faceted nature of adversarial machine learning, where the interplay between attack and defense mechanisms continues to evolve. Finally, Table 1 provides a categorization of the characteristics of the above APAs, and the visual examples provided in Figure 1 highlights the visual differences across availability poisoning attacks.

## 2.2 Availability Poisoning Defenses

The goal of the defender is to successfully train a model with good generalization abilities (*e.g.*, test accuracies on natural unseen images) on protected data. Generally, the defender can control the training algorithm, and only have

Table 1: Availability poisoning attack algorithms implemented in APBench. "Type" and "Budget" respectively denotes the type of perturbation and its budget. "Mode" denotes the training mode, where "S" and "U" and respectively mean supervised and unsupervised training. "No surrogate" denotes whether the attack requires access to a surrogate model for perturbation generation. "Class-wise" and "Sample-wise" indicate if the attack supports class-wise and sample-wise perturbation generation. "Stealthy" denotes whether the attack is stealthy to human.

| Attack Method | Type | Budget | Mode | No surrogate | Class-wise | Sample-wise | Stealthy |
|---|---|---|---|---|---|---|---|
| DC (Feng et al., 2019) | | | S | | | ✓ | ✓ |
| NTGA (Yuan & Wu, 2021) | | | S | | | ✓ | ✓ |
| HYPO (Tao et al., 2021) | | | S | | | ✓ | ✓ |
| EM (Huang et al., 2021) | $\ell_\infty$ | 8/255 | S | | ✓ | ✓ | ✓ |
| REM (Fu et al., 2022) | | | S | | ✓ | ✓ | ✓ |
| TAP (Fowl et al., 2021) | | | S | | | ✓ | ✓ |
| UCL (He et al., 2023) | | | U | | ✓ | ✓ | ✓ |
| TUE (Ren et al., 2023) | | | U | | | ✓ | ✓ |
| LSP (Yu et al., 2022) | $\ell_2$ | 1.30 | S | ✓ | ⊙ | ✓ | |
| AR (Sandoval-Segura et al., 2022b) | | 1.00 | S | ✓ | ⊙ | ✓ | ✓ |
| OPS (Wu et al., 2023) | $\ell_0$ | 1 | S | ✓ | ✓ | | |

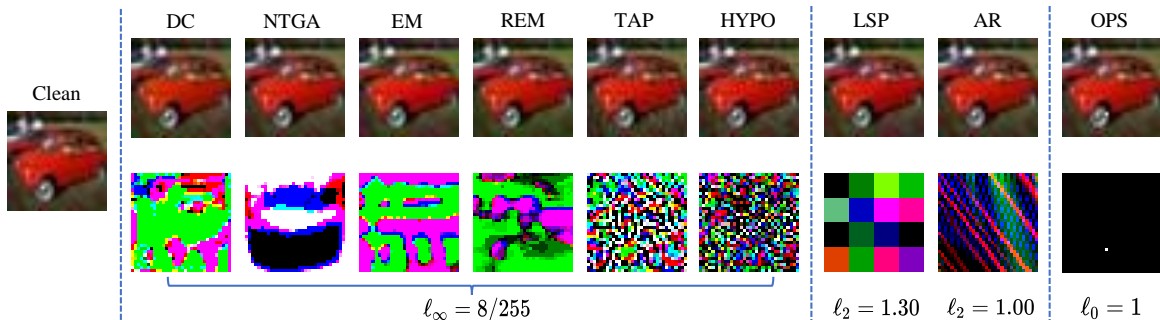

Figure 1: Visualizations of unlearnable CIFAR-10 images with corresponding perturbations. Perturbations are normalized for visualization.

access to a training data set with data poisoned either partially or fully. The objective of the defender is thus to find a novel training algorithm $g(\mathcal{D}_{\text{poi}})$ that trains models to generalize well to the original data distribution:

$$\min_g \mathbb{E}_{(\mathbf{x}_i, y_i) \sim \mathcal{D}_{\text{test}}}[\mathcal{L}(f_{\boldsymbol{\theta}^\star}(\mathbf{x}_i), y_i)], \text{ s.t. } \boldsymbol{\theta}^\star = g(\mathcal{D}_{\text{poi}}). \tag{2}$$

Notably, if the method employs the standard training loss but performs novel image transformations $h$, then $g$ can be further specialized as follows:

$$g(\mathcal{D}_{\text{poi}}) = \text{argmin}_{\boldsymbol{\theta}} \, \mathbb{E}_{(\hat{\mathbf{x}}_i, y_i) \sim \mathcal{D}_{\text{poi}}(\boldsymbol{\delta})} \, \mathcal{L}(f_{\boldsymbol{\theta}}(h(\hat{\mathbf{x}}_i)), y_i). \tag{3}$$

Currently, defense methods against availability poisoning can be mainly classified into two categories: preprocessing and training-phase defenses. Data preprocessing methods preprocess the training images to eliminate the poisoning perturbations prior to training. Here, we summarize the defense methods implemented in APBench, and Table 2 categorizes the above defense methods in terms of their time costs, types, and descriptions:

- **Adversarial training (AT)** (Madry et al., 2017): AT is a widely-recognized effective approach against availability poisoning Huang et al. (2021); Fu et al. (2022). Small error-maximizing adversarial perturbations are applied to the training images during training, in order to improve the robustness of the model against APA perturbations.

- **Image Shortcut Squeezing (ISS)** (Liu et al., 2023): ISS uses traditional image compression techniques such as grayscale transformation, bit-depth reduction (BDR), and JPEG compression, as defenses against availability poisoning.

- **Early stopping (ES)** (Sandoval-Segura et al., 2022a): As training on availability poisons typically leads to early overfitting on availability poisons, early stopping, *i.e.*, stopping the training process when the model reaches peak validation accuracy, may be practical as a defense mechanism.

- **Adversarial augmentations (UEraser-Lite, UEraser-Max)** (Qin et al., 2023): APBench implements two flavors of UEraser. First, similar to ISS, UEraser-Lite uses an effective augmentation pipeline to suppress availability poisoning shortcuts. UEraser-Max further improves the defense against availability poisoning with error-maximizing adversarial augmentations. Specifically, UEraser-Max samples multiple augmentations per image, and train models on the augmented images with maximum loss to prevent learning from poisoning shortcuts.

- **AVATAR** (Dolatabadi et al., 2023): Inspired by DiffPure (Nie et al., 2022) which improves adversarial robustness by using diffusion models to remove adversarial perturbations, AVATAR similarly cleans the images of the unlearnable perturbations with diffusion models.

- **Standard data augmentations**: For referential baselines, APBench also includes commonly used data augmentation techniques such as Gaussian blur, random crop and flip (standard training), CutOut (DeVries & Taylor, 2017), CutMix (Yun et al., 2019), and MixUp (Zhang et al., 2018), and show their (limited) effect in mitigating availability poisons.

Table 2: Availability poisoning defense algorithms implemented in APBench.

| Defense Method | Type | Time Cost | Description |
|---|---|---|---|
| Standard | Data augmentations | Low | Random image cropping and flipping |
| CutOut (DeVries & Taylor, 2017) | | Low | Random image erasing |
| CutMix (Yun et al., 2019) | | Low | Random image cutting and stitching |
| MixUp (Zhang et al., 2018) | | Low | Random image blending |
| Gaussian (used in (Liu et al., 2023)) | Data preprocessing | Low | Image Gaussian blurring |
| BDR (used in (Liu et al., 2023)) | | Low | Image bit-depth reduction |
| Grayscale (used in (Liu et al., 2023)) | | Low | Image grayscale transformation |
| JPEG (used in (Liu et al., 2023)) | | Low | Image compression |
| AVATAR (Dolatabadi et al., 2023) | | High | Image corruption and restoration |
| Early stopping (Sandoval-Segura et al., 2022a) | Training-phase defense | Low | Finding peak validation accuracy |
| UEraser-Lite (Qin et al., 2023) | | Low | Stronger data augmentations |
| UEraser-Max (Qin et al., 2023) | | High | Adversarial augmentations |
| AT (Madry et al., 2017) | | High | Adversarial training |

## 2.3 Related Benchmarks

Availability poisoning is closely connected to the domains of *adversarial* and *backdoor* attack and defense algorithms. Adversarial attacks primarily aim to deceive models with adversarial perturbations during inference to induce misclassifications. There are several libraries and benchmarks available for evaluating adversarial attack and defense techniques, such as Foolbox (Rauber et al., 2020), AdvBox (Goodman et al., 2020), and RobustBench (Croce et al., 2020).

*Backdoor* or *data poisoning* attacks (Cinà et al., 2023) focus on injecting backdoor triggers into the training algorithm or data respectively, causing trained models to misclassify images containing these triggers while maintaining or minimally impacting clean accuracy. Benchmark libraries specifically designed for backdoor attacks and defenses include TrojanZoo (Pang et al., 2020), Backdoorbench (Wu et al., 2022), and Backdoorbox (Li et al., 2023). Moreover, Schwarzschild et al. (2021) also introduced benchmarks for data poisoning and backdoor attacks, which explores their ability to generalize to different testing settings. Interestingly, it examines the "triggerless" attack scenario, where it forces the model to misclassify a *specific* target image, opening a backdoor in the model for post-training attack

opportunities. In contrast, attacks in APBench aims to prevent the model from learning from features present in the *entire* poisoned set of images to preserve user privacy. Geiping et al. (2021) extends the benchmark to include additional data poisoning attacks. To summarize, the above benchmark suites all examine **malicious attacks in order to control the model's behavior**, whereas the goal of APAs considered in APBench is benign, aiming to protect individual privacy by perturbing their own personal data.

There is currently a lack and an urgent need of a dedicated and comprehensive benchmark that standardizes and evaluates availability poisoning attack and defense strategies. To the best of our knowledge, APBench is the first benchmark that fulfills this purpose. It offers an extensive library of recent attacks and defenses, explores various perspectives, including the impact of poisoning rates and model architectures, as well as attack transferability. We hope that APBench can make significant contributions to the community and foster the development of future availability attacks for effective privacy protection.

## 3  A Unified Availability Poisoning Benchmark

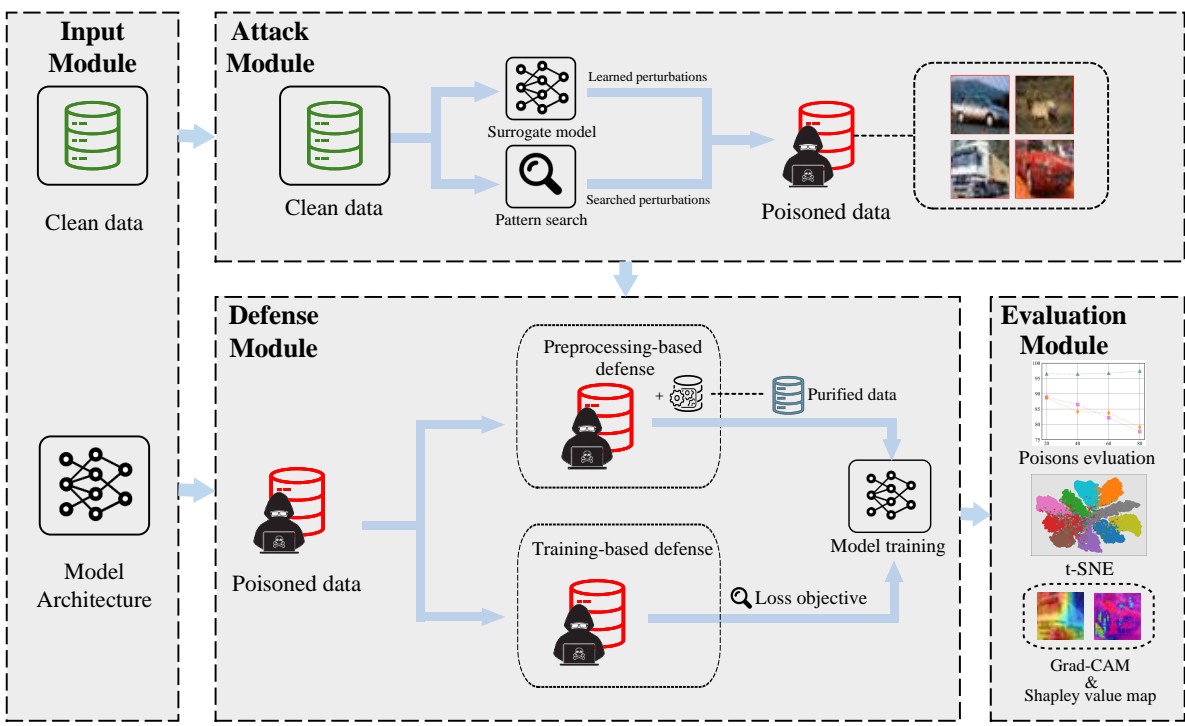

Figure 2: The overall system design of APBench.

As shown in Figure 2, APBench consists of three main components: (a) The availability poisoning attack module. This library includes a set of representative availability poisoning attacks that can generate unlearnable versions of a given clean dataset. (b) The poisoning defense module. This module integrates a suite of state-of-the-art defenses that focus on mitigating the unlearning effect and restore clean accuracies to a certain extent. (c) The evaluation module. This module can efficiently analyze the performance of various availability poisoning attack methods using accuracy metrics and visual analysis strategies.

We built an extensible codebase as the foundation of APBench. In the attack module, we provide a total of 9 availability poisoning attacks of 3 different perturbation types ($\ell_p$) for supervised learning, and 2 attacks for unsupervised learning. For each availability poisoning attack method, we can generate their respective poisoned datasets. This module also allows us to expand to different perturbations budgets and poisoning ratios, and easily extend to future poisoning methods. Using the poisoned datasets generated by the attack module, we can evaluate defenses through the defense module, which currently includes 8 defense algorithms and 4 conventional data augmentation methods. The goal of this module is to ensure that models trained on unlearnable datasets can still generalize well on clean data. The defense

module primarily achieves poisoning mitigation through data preprocessing or training-phase defenses. Finally, the evaluation module computes the accuracy metrics of different attacks and defense combinations, and can also perform qualitative visual analyses to help understand the characteristics of the datasets.

# 4 Evaluations

## 4.1 Setups

**Datasets** We evaluated our benchmark on 4 commonly datasets (CIFAR-10 (Krizhevsky et al., 2009), CIFAR-100 (Krizhevsky et al., 2009), SVHN (Netzer et al., 2011), and an ImageNet (Deng et al., 2009) subset) and 7 mainstream models, comprising 5 CNN-based models (ResNet-18 (He et al., 2016), ResNet-50 (He et al., 2016), MobileNetV2 (Sandler et al., 2018), and DenseNet-121 (Huang et al., 2017)) and 2 ViT models (ViT-small (Dosovitskiy et al., 2021) and CaiT-small (Touvron et al., 2021)). To ensure a fair comparison between all methods, we evaluate the methods on the general and standardized model architectures. Appendix A further describes the detail specifications of the datasets and the respective test accuracies achievable through standard training on clean training data.

**Attacks and defenses** For a complete introduction of the APA attacks and defense methods, please refer to Section 2.

**Types of Threat Models** We classify APAs based on three distinct perturbation types: $\ell_\infty$-bounded attacks (DC, NTGA, EM, REM, TAP, and HYPO); $\ell_2$-bounded attacks (LSP and AR); an $\ell_0$-bounded attack (OPS). Notably, the $\ell_0$-bounded OPS attack demonstrates remarkable resilience against a wide range of defenses, resisting disruption from image preprocessing or augmentations and remaining unaffected by $\ell_\infty$ adversarial training. Conversely, in terms of stealthiness, the $\ell_0$ attacks are less subtle than their $\ell_\infty$ and $\ell_2$ counterparts, as illustrated in Figure 1. Visually, perturbations bounded by both $\ell_\infty$ and $\ell_2$ are comparable *w.r.t.* the degree of visual stealthiness and effectiveness. Finally, as the two $\ell_2$-bounded attacks (LSP and AR) do not require surrogate model training, they are thus more efficient in the unlearnable examples synthesis.

**Training settings** We trained the CIAFR-10, CIFAR-100 and ImageNet-subset models for 200 epochs and the SVHN models for 100 epochs. We used the stochastic gradient descent (SGD) optimizer with a momentum of 0.9 and a learning rate of 0.1 by default. As for unsupervised learning, all experiments are trained for 500 epochs with the SGD optimizer. The learning rate is 0.5 for SimCLR (Chen et al., 2020a) and 0.3 for MoCo-v2 (Chen et al., 2020b). Please note that we generate sample-wise perturbations for all APAs. Specific settings for each defense method may have slight differences, and detailed information can be found in the Appendix A.

**Evaluation objectives** In this section, we first evaluated the effectiveness of APA attacks against the defense methods (Section 4.2). Moreover, we explored challenging scenarios with real-world constraints, such as varying data poisoning rates and perturbation magnitudes on different models, assessed the transferability of attacks across different model architectures (Section 4.3), and APAs under unsupervised learning (UCL (He et al., 2023) and TUE (Ren et al., 2023)). The implementation details of the algorithms and additional results can be found in Appendix A and Appendix B respectively. Below, we present the key takeaways from our evaluations:

---

**Key Takeaways**

1. The efficacy of availability poisoning attacks in terms of hindering the model's generalization ability is suboptimal when the poisoning rate is relatively low. However, it can still ensure the effectiveness of privacy protection for data that has been poisoned, when no defense is employed.

2. Nearly all availability poisoning attacks exhibit a high degree of transferability across diverse model architectures, with the exception of AR, which appears to be ineffective on CaiT-small. This suggests that current attacks may exploit common training algorithmic flaws rather than model-specific vulnerabilities. Furthermore, all defenses demonstrate similar defense performance on different architectures.

3. For defenses, the most promising method is based on pretrained diffusion generative models, *e.g.*, AVATAR. Without the availability of additional data or pretrained models, UEraser variants stand out as the most effective defense methods.

---

## 4.2 Standard Scenario

To start, we consider a common scenario where both the surrogate model and target model are ResNet-18, and the poisoning rate is set to $100\%$. We first evaluate the performance of the supervised poisoning methods against the defense mechanisms and commonly used data augmentation strategies. Table 3 and Table 4 present the evaluation results on CIFAR-10 from our benchmark.

It is evident that the conventional data augmentation methods appear to be ineffective against all poisoning methods. Yet, even simple image compression methods (BDR, grayscale, and JPEG corruption from ISS (Liu et al., 2023)) demonstrate a notable effect in mitigating the poisoning attacks, but fails to achieve high clean accuracy.

Despite requiring more computational cost or additional resources (pretrained diffusion models for AVATAR), methods such as UEraser-Max (Qin et al., 2023) and AVATAR (Dolatabadi et al., 2023), generally surpass the image compression methods from ISS in terms of effectiveness. While AVATAR is inferior to UEraser-Max in gaining accuracy, it decouples the defense into an independent data sanitization phase, making it more universally applicable, but requiring a pretrained diffusion model. While the early stopping (ES) method can be somewhat effective as a defense, is not usually considered a good one, with subpar peak accuracies. Adversarial training appears effective but in many cases is outperformed by JPEG compression. Notably, it also fails against OPS, as the $\ell_\infty$ perturbation budget cannot mitigate $\ell_0$ threats.

Table 3: Test accuracies (%) of models trained on poisoned CIFAR-10 datasets. The model trained on a clean CIFAR-10 dataset attains an accuracy of $94.32\%$. We highlight best defenses in bold (highest accuracy for the same attack), and best attacks with underline (lowest accuracy for the same defense). "ND" – No Defense, "ST" – Standard training, "CO" – CutOut, "CM" – CutMix, "MU" – MixUp, "Gauss." – Gaussian, "BDR" – Bit-Depth Reduction, "GS" – Grayscale, "JPEG" – JPEG Compression, "AVA" – AVATAR, "ES" – Early Stopping, "U-Max" – UEraser-Max, and "AT" – adversarial training.

| Method | ND | ST | CO | CM | MU | Gauss. | BDR | GS | JPEG | AVA | ES | U-Max | AT |
|---|---|---|---|---|---|---|---|---|---|---|---|---|---|
| DC | 14.17 | 15.19 | 19.94 | 17.91 | 25.07 | 16.10 | 67.73 | 85.55 | 83.57 | 82.10 | 26.08 | **92.17** | 76.85 |
| EM | 19.68 | 20.78 | 18.79 | 22.28 | 31.14 | 14.71 | 37.94 | 92.03 | 80.72 | 75.62 | 25.39 | **93.61** | 82.51 |
| REM | 17.81 | 17.47 | 21.96 | 26.22 | 43.07 | 21.80 | 58.60 | 92.27 | 85.44 | 82.42 | 31.32 | **92.43** | 77.46 |
| HYPO | 69.04 | 70.38 | 69.04 | 67.12 | 74.25 | 62.17 | 74.82 | 63.35 | 85.21 | 85.94 | 70.52 | **88.44** | 81.49 |
| NTGA | 17.82 | 22.76 | 13.78 | 12.91 | 20.59 | 19.95 | 59.32 | 70.41 | 68.72 | 86.22 | 28.19 | **86.78** | 69.70 |
| TAP | 4.34 | 6.27 | 9.88 | 14.21 | 15.46 | 7.88 | 70.75 | 11.01 | 84.08 | **87.75** | 39.54 | 79.05 | 79.92 |
| LSP | 10.19 | 13.06 | 14.96 | 17.69 | 18.77 | 18.61 | 53.86 | 64.70 | 80.14 | 76.90 | 29.10 | **92.83** | 81.38 |
| AR | 12.63 | 11.74 | 10.95 | 12.60 | 14.15 | 13.83 | 36.14 | 35.17 | 84.75 | 88.60 | 44.29 | **90.12** | 81.15 |
| OPS | 11.94 | 14.69 | 52.98 | 64.72 | 49.27 | 13.38 | 37.32 | 19.88 | **78.48** | 66.16 | 38.20 | 77.99 | 14.95 |

Table 4: Test accuracies (%) of ViT-small models trained on poisoned CIFAR-10 datasets. The model trained from scratch on a clean CIFAR-10 dataset attains an accuracy of $84.66\%$. Value highlight rules and method abbreviations are the same as in Table 3.

| Method | ND | ST | CO | CM | MU | Gauss. | BDR | GS | JPEG | AVA | ES | U-Max | AT |
|---|---|---|---|---|---|---|---|---|---|---|---|---|---|
| DC | 20.37 | 22.94 | 26.03 | 32.80 | 33.64 | 26.06 | 55.47 | 72.45 | 71.82 | 79.16 | 26.49 | **82.89** | 77.26 |
| EM | 36.82 | 37.14 | 38.79 | 40.30 | 46.91 | 36.29 | 54.59 | 59.38 | 70.69 | 72.81 | 41.29 | **81.64** | 78.03 |
| REM | 30.61 | 31.72 | 29.90 | 34.68 | 44.99 | 33.14 | 50.59 | 80.29 | 74.63 | 75.52 | 42.88 | **82.50** | 79.32 |
| HYPO | 74.47 | 75.23 | 75.76 | 74.60 | 76.16 | 75.12 | 79.21 | 63.62 | 75.77 | 83.15 | 67.29 | **83.50** | 78.62 |
| TAP | 22.38 | 22.93 | 22.65 | 24.86 | 27.82 | 23.47 | 40.03 | 37.75 | 67.26 | 73.26 | 31.17 | 69.83 | **76.55** |
| NTGA | 26.40 | 28.27 | 22.14 | 23.39 | 39.58 | 25.57 | 52.38 | 56.92 | 53.94 | 74.88 | 33.26 | 76.39 | **77.08** |
| LSP | 27.14 | 29.06 | 30.17 | 32.66 | 28.38 | 33.07 | 41.17 | 59.34 | 68.07 | 66.74 | 32.69 | **87.01** | 76.94 |
| AR | 22.37 | 25.04 | 26.92 | 21.18 | 30.67 | 25.48 | 37.04 | 38.90 | 74.77 | 78.64 | 45.54 | 63.90 | **75.62** |
| OPS | 18.16 | 20.84 | 61.27 | **76.59** | 34.58 | 32.39 | 45.71 | 31.60 | 69.31 | 59.26 | 22.63 | 66.72 | 24.30 |

We further conduct experiments representative poisoning methods (EM, REM, LSP and AR) on the CIFAR-100, SVHN, and ImageNet-subset datasets, and the results are shown in Table 5.

Table 5: Test accuracies (%) on poisoned CIFAR-100, SVHN and ImageNet-subset (IN-100) datasets. Value highlight rules and method abbreviations are the same as in Table 3.

| Dataset | Method | ND | ST | CO | CM | MU | Gauss. | BDR | GS | JPEG | ES | U-Max |
|---------|--------|-----|-----|-----|-----|-----|--------|------|------|------|------|-------|
| CIFAR-100 | EM | 2.94 | 3.03 | 4.15 | 3.98 | 6.46 | 2.99 | 34.10 | 59.14 | 58.71 | _7.06_ | **68.81** |
| | REM | 3.57 | 3.73 | 4.00 | 3.71 | 10.90 | 3.59 | 29.16 | 57.47 | 55.60 | 10.99 | **67.72** |
| | LSP | 2.39 | 2.56 | 2.33 | 4.52 | 4.86 | _1.71_ | _27.12_ | 39.45 | _52.82_ | 9.52 | **68.31** |
| | AR | _2.04_ | _1.87_ | _1.63_ | _3.17_ | _2.35_ | 2.62 | 31.15 | _16.13_ | 54.73 | 26.58 | **55.95** |
| SVHN | EM | _11.07_ | _10.33_ | 13.38 | 10.77 | _12.79_ | 8.82 | 36.65 | 65.66 | 86.14 | 13.47 | **90.24** |
| | REM | 12.95 | 14.02 | 18.92 | 9.55 | 19.56 | 7.54 | 42.52 | 19.59 | **90.58** | 19.61 | _88.26_ |
| | LSP | 11.38 | 12.16 | _12.98_ | 8.17 | 18.86 | _7.15_ | _26.67_ | 16.90 | _84.06_ | _12.91_ | **90.64** |
| | AR | 17.69 | 19.23 | 14.92 | _6.71_ | 13.52 | 7.75 | 39.24 | _10.00_ | **92.46** | 89.32 | 90.07 |
| IN-100 | EM | _1.90_ | _2.94_ | _4.05_ | _4.73_ | 4.15 | _3.15_ | 6.45 | 12.20 | _31.73_ | _8.80_ | **44.07** |
| | REM | 3.82 | 3.66 | 4.13 | 4.78 | _3.94_ | 4.28 | _4.03_ | _3.95_ | 40.98 | 17.19 | _42.14_ |
| | LSP | 30.64 | 38.52 | 40.56 | 29.78 | 7.85 | 42.68 | 26.58 | 25.18 | 36.83 | 39.52 | **63.28** |

To summarize, our findings indicate that perturbations constrained by traditional $\ell_p$ norms are ineffective against adversarial augmentation (UEraser-Max), and image restoration by pretrained diffusion models (AVATAR), as they break free from the assumption of $\ell_p$ constraints. Even simple image compression techniques (JPEG, Grayscale, and BDR) can effectively remove the effect of perturbations. At this moment, APAs that rely on $\ell_p$-bounded perturbations may not be as effective as initially suggested by the relevant publications.

## 4.3 Challenging Scenarios

To further investigate the effectiveness and robustness of APA attacks and defenses, we conducted evaluations in more challenging scenarios with real-world constraints. Namely, we considered scenarios include partial poisoning, larger perturbation budgets, and the attack transferability to different models.

**Privacy protection under partial poisoning** In realistic scenarios, it is difficult for an attacker to achieve modification of the entire dataset. We thus investigate the impact of partial poisoning rates, where only a portion of the training data is perturbed by APAs. This raises the following question: *Are APAs effective in protecting only a portion of the training data?* To answer, we introduce poisoning perturbations with APAs to a varying portion of the training data, and investigate how well the models learn the origin features that exist in the poisoned images for different poisoning rates. Specifically, we explore 4 different poisoning rates (20%, 40%, 60%, 80%). Figure 4 evaluates and compares the mean losses of the unlearnable images used during training ("Unlearnable"), the origin images of the unlearnable part ("Clean"), and for reference, the mean losses of images unseen by the model from the test set ("Test"), and "Train" means the loss of the clean part of the training set. For a similar view on the accuracies, please refer to Figure 3. We find that the losses on the original images of the unlearnable part is similar to that of the test set, or even lower. This suggests that APAs can *reasonably protect the private data against undefended learning*. For results under defenses, please refer to Appendix B.

**Larger perturbations** We increased the magnitude of APA perturbations to further evaluate the performance of attacks and defenses. Table 6 presents the results of APAs with larger perturbations on CIFAR-10. We note that larger perturbations indeed have a more pronounced impact on suppressing defense performance, leading to significant accuracy losses for all defense methods. There exists a trade-off between perturbation magnitude and accuracy recovery. Considering that at larger perturbations, APAs generate images that are perceptually of lower quality, yet some defense methods are still effective, which may make such attacks less practical in real-world scenarios.

**Attack transferability across models** In real-world scenarios, APAs can only manipulate the data and do not have access to specific details of the defender. Therefore, we evaluated the transferability of APAs by using ResNet-18 as the surrogate for the surrogate-based attacks, and transfer the perturbed data to be trained on other models, including 4 CNN-based models (ResNet-50, SENet-18, MobileNetV2, DenseNet-121), and 4 ViT models (ViT-small, and

Table 6: Test accuracies (%) on poisoned CIFAR-10 datasets with increased perturbations. We only highlight the best defenses per attack in bold, as the attacks may have different perturbation budgets.

| Method | Type | Budget | Standard | Grayscale | JPEG | ES | UEraser-Max | AT |
|--------|------|--------|----------|-----------|------|-----|-------------|-----|
| EM | $\ell_\infty$ | 8/255 | 20.78 | 92.03 | 80.72 | 25.39 | **93.61** | 82.51 |
|    |      | 16/255 | 18.74 | 76.76 | 55.96 | 27.39 | **88.09** | 77.82 |
| REM | $\ell_\infty$ | 8/255 | 17.47 | 92.27 | 85.44 | 31.32 | **92.43** | 77.46 |
|     |      | 16/255 | 19.80 | **83.65** | 80.07 | 33.07 | 80.36 | 75.64 |
| LSP | $\ell_2$ | 1.30 | 13.06 | 64.70 | 80.14 | 29.10 | **92.83** | 81.38 |
|     |      | 1.74 | 15.83 | 37.60 | 42.83 | 27.30 | **87.20** | 77.92 |
| AR | $\ell_2$ | 1.00 | 11.74 | 35.17 | 84.75 | 44.29 | **90.12** | 81.15 |
|    |      | 1.50 | 11.20 | 26.10 | **78.24** | 20.96 | 68.42 | 70.14 |

CaiT-small). The results are shown in Table 7. It is evident that all surrogate-based and -free poisoning methods exhibit strong transferability, while the three recently proposed defenses also achieve successful defense across different model architectures. The only exception is the AR method, which fails against CaiT-small.

**Adaptive poisoning** We evaluated strong adaptive poisons against various defenses using two poisoning methods, EM (Huang et al., 2021) and REM (Fu et al., 2022). We assume that the defenders can be adapted to three defenses (Grayscale, JPEG, and UEraser), by using the attack in the perturbation generation process. From Tables 8 and 9, it can be seen that adaptive poisoning significantly affects the performance of the Grayscale defense, but has less effect on JPEG and UEraser.

**Unsupervised learning** We evaluated the APAs targeting unsupervised models on CIFAR-10. We considered 2 popular unsupervised learning algorithms: SimCLR (Chen et al., 2020a) and MoCo-v2 (Chen et al., 2020b). All defense methods were applied by preprocessing images prior to the unsupervised learning. Therefore, we used UEraser-Lite as a data preprocessing method by sampling the augmented images instead of as a training-phase defense. The results of all experiments are shown in Table 10.

## 4.4 Other Evaluations

**Adversarial robustness of trained models** Table 11 additionally compared the robustness of the trained models on CIFAR-10 under the projected gradient descent (PGD) adversarial attack with a perturbation budget of $8/255$. Note that all experiments considered both scenario including white-box attacks and black-box transferability from a similarly-trained model with a different initialization. The PGD attack used 20 steps with a step size of $2/255$. As

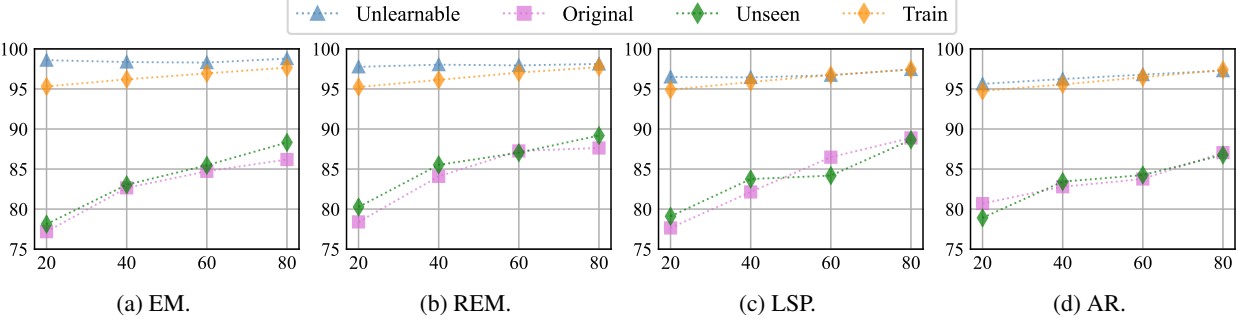

Figure 3: The accuracies (%, vertical axes) indicate that original features in unlearnable examples are not learned by the model. All evaluations consider partial poisoning scenarios (poisoning rates from 20% to 80%, horizontal axes). Note that "Unlearnable" and "Original" respectively denote the set of unlearnable examples and their original clean variants, and "Unseen" denote images from the test set unobserved during model training.

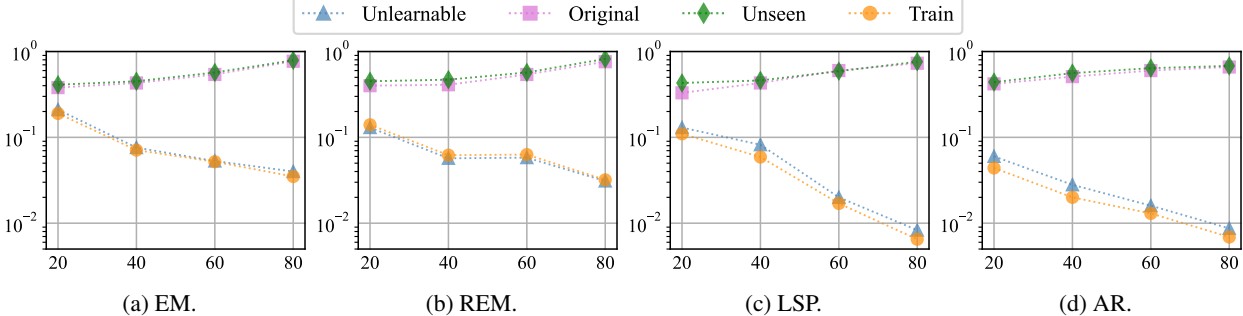

Figure 4: The mean losses (vertical axes) indicate that original features in unlearnable examples are not learned by the model. The evaluation setting follows Figure 3.

Table 7: Clean test accuracies of different CIFAR-10 target models, where attacks are oblivious to the model architectures. Note that AR and LSP are surrogate-free, and for EM and REM the surrogate model is ResNet-18.

| Model | Clean | Method | Standard | Grayscale | JPEG | ES | UEraser-Max | AVATAR |
|-------|-------|--------|----------|-----------|------|-----|-------------|--------|
| ResNet-50 | 94.47 | EM | 14.41 | 83.40 | 76.88 | 26.69 | **85.89** | 77.64 |
| | | REM | 16.26 | 87.26 | 75.79 | 31.37 | **92.69** | 83.68 |
| | | LSP | 19.23 | 68.94 | 73.24 | 32.73 | **93.08** | 76.47 |
| | | AR | 11.83 | 27.51 | 80.24 | 28.66 | 81.40 | **86.39** |
| SENet-18 | 94.83 | EM | 13.60 | **86.03** | 79.35 | 16.35 | 83.27 | 74.22 |
| | | REM | 20.99 | 84.50 | 78.92 | 22.85 | **93.17** | 84.37 |
| | | LSP | 18.54 | 65.06 | 76.51 | 26.38 | **92.53** | 75.19 |
| | | AR | 13.68 | 34.26 | 79.29 | 37.04 | 75.06 | **84.37** |
| MobileNetV2 | 94.62 | EM | 15.62 | 77.21 | 70.96 | 16.71 | **82.71** | 75.62 |
| | | REM | 20.83 | 80.81 | 72.27 | 21.92 | **91.03** | 82.77 |
| | | LSP | 16.82 | 61.07 | 72.03 | 28.12 | **92.10** | 76.81 |
| | | AR | 13.36 | 28.54 | 68.14 | 39.45 | 73.40 | **81.63** |
| DenseNet-121 | 95.08 | EM | 13.89 | **82.49** | 78.42 | 15.68 | 82.37 | 76.69 |
| | | REM | 21.45 | 85.47 | 78.42 | 22.35 | **93.09** | 83.04 |
| | | LSP | 18.94 | 67.95 | 74.90 | 26.86 | **93.47** | 78.22 |
| | | AR | 13.43 | 25.51 | 81.12 | 36.51 | 82.36 | **89.92** |
| ViT-small | 84.66 | EM | 21.47 | **80.42** | 72.64 | 30.91 | 74.29 | 54.84 |
| | | REM | 32.17 | 79.65 | 74.92 | 43.07 | **83.27** | 73.57 |
| | | LSP | 29.06 | 59.34 | 68.07 | 32.69 | **87.01** | 66.74 |
| | | AR | 25.04 | 38.90 | 74.77 | 45.54 | 63.90 | **78.64** |
| CaiT-small | 71.96 | EM | 17.01 | **64.76** | 63.75 | 39.69 | 63.37 | 41.94 |
| | | REM | 26.11 | 65.05 | 66.43 | 47.39 | **72.05** | 62.53 |
| | | LSP | 25.08 | 63.06 | 57.15 | 37.95 | **70.92** | 51.39 |
| | | AR | 68.63 | 66.27 | 69.30 | 67.41 | **70.04** | 62.77 |

Table 8: Test accuracies (%) of adaptive poisoning with EM on ResNet-18.

| EM + | Standard | Grayscale | JPEG | U-Max |
|------|----------|-----------|------|-------|
| Grayscale | 19.48 | 21.64 | 78.39 | **90.52** |
| JPEG | 20.67 | 90.29 | 76.25 | **93.22** |
| UEraser | 35.24 | 88.62 | 80.46 | **89.55** |

Table 9: Test accuracies (%) of adaptive poisoning with REM on ResNet-18.

| REM + | Standard | Grayscale | JPEG | U-Max |
|-------|----------|-----------|------|-------|
| Grayscale | 16.70 | 56.33 | 82.47 | **91.37** |
| JPEG | 19.45 | 91.71 | 75.84 | **92.53** |
| UEraser | 21.61 | 89.26 | 77.51 | **91.84** |

Table 10: Performance of APA attacks and defenses on different unsupervised learning algorithms and datasets. Note that "U-Lite" denotes UEraser-Lite.

| Algorithm | Method | Standard | Grayscale | JPEG | AVATAR | UEraser-Lite |
|---|---|---|---|---|---|---|
| SimCLR | UCL | 47.25 | 46.91 | 66.76 | **83.22** | 68.42 |
| | TUE | 57.10 | 56.37 | 67.54 | **84.24** | 66.59 |
| MoCo-v2 | UCL | 53.78 | 53.34 | 65.44 | **83.08** | 72.13 |
| | TUE | 66.73 | 64.95 | 67.28 | **82.48** | 74.82 |

Table 11: Test accuracies (%) on CIFAR-10 PGD-20 adversarial examples.

| Type | Method | Standard | Grayscale | JPEG | AVATAR | UEraser-Max | AT |
|---|---|---|---|---|---|---|---|
| Black-box | EM | 16.83 | 19.77 | **73.85** | 20.07 | 71.43 | 69.77 |
| | REM | 22.53 | 19.16 | 73.70 | 17.36 | **80.46** | 70.74 |
| | LSP | 14.29 | 24.34 | 67.01 | 15.42 | **80.40** | 71.39 |
| | AR | 15.82 | 16.55 | 75.21 | 16.84 | **76.97** | 68.81 |
| White-box | EM | 0.00 | 0.00 | 3.74 | 0.00 | 1.37 | **28.60** |
| | REM | 0.00 | 0.00 | 3.35 | 0.00 | 3.11 | **32.52** |
| | LSP | 0.00 | 0.00 | 0.39 | 0.00 | 3.83 | **34.45** |
| | AR | 0.00 | 0.00 | 2.59 | 0.00 | 2.83 | **32.04** |

expected, adversarial training provides the best defense against white-box attacks. While standard training, Grayscale and AVATAR failed to gain adversarial robustness, interestingly, JPEG compression and UEraser-Max demonstrated small improvements against white-box attacks, and notable robustness against black-box attacks, even surpassing that of adversarial training.

**Visual analyses** We provide visualization tools (Grad-CAM (Selvaraju et al., 2017) and Shapley Value Maps (Lundberg & Lee, 2017)) to facilitate the analysis and understanding of APAs. We also use t-SNE (Van der Maaten & Hinton, 2008) to visualize the availability poisons (Figure 7). Although t-SNE cannot accurately represent high-dimensional spaces, it aids in the global visualization of feature representations, allowing us to observe specific characteristics of availability poisons. For additional discussions on the visualizations, please refer to Appendix B.1.

# 5 Discussions

**Future outlook of APAs** Future research directions on APAs should explore methods that enhance the resilience of perturbations. Promising future directions of APAs include *generalizable attacks*, which can simultaneously target the DNNs being trained, leveraging diffusion models for stronger attacks, and how to remain robust against *color distortions*-based defenses. Moreover, semantic-based perturbations that breaks away from the constraints of $\ell_p$ norms may offer an alternative strategy to enhance the effectiveness of APAs while preserving the perceptual quality of images. Such modifications to images can be challenging to remove by a wide-range of defenses.

**Limitations** APBench has mainly focused on providing algorithms and evaluations related to image data. However, such availability poisoning methods may also be applicable to text, speech, or video domains. In the future, we plan to expand APBench to include more domains, aiming to establish a more comprehensive benchmark for personal privacy protection across various modalities against deep learning.

**Reproducibility Statement** We provide an open-source implementation of all attacks and defenses in the supplementary material. Following the README file, users should be able to run all experiments on their own machines to reproduce the results shown in paper.

**Ethics Statement** Similar to many other technologies, the implementation of APAs can be used for both beneficial and malicious purposes. We understand that APAs were originally proposed to protect privacy, but they can also be used to generate maliciously data to introduce model backdoors. The benchmark aims to promote an understanding of

various APA attack and defense methods, as well as encourage the development of new algorithms in this field. It is also important for us to raise awareness of the false sense of security provided by APAs However, we emphasize that the use of these algorithms and evaluation results should comply with ethical guidelines and legal regulations. We encourage users to be aware of the potential risks of the technology and take appropriate measures to ensure its beneficial use for both society and individuals.

## 6 Conclusions

We have established the first comprehensive and up-to-date benchmark for the field of availability poisoning, covering a diverse range of availability poisoning attacks and state-of-the-art defense algorithms. We have conducted evaluations and analyses of different combinations of attacks and defenses, as well as additional challenging scenarios. Through this new benchmark, our primary objective is to provide researchers with an in-depth empirical and comparative analysis of the current progress in the field of availability poisoning attacks and defenses. We hope it can enable rapid comparisons between existing methods and new approaches, while also inspiring fresh ideas through our comprehensive benchmark and analysis tools. We believe that our benchmark will contribute to the advancement of availability poisoning research and the development of more effective attacks to safeguard privacy in the future.

## Acknowledgements

This work is supported in part by National Natural Science Foundation of China (Nos. 62376263 and 62372443), Guangdong Basic and Applied Basic Research Foundation (Nos. 2023B1515130002), Natural Science Foundation of Guangdong (Nos. 2024A1515030209 and 2024A1515011970), and Shenzhen Science and Technology Innovation Commission (Nos. JCYJ20230807140507015 and JCYJ20220531100804009).

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

## A  Experimental Settings

### A.1  Datasets

Table 12 summarizes the specifications of datasets and the respective test accuracies on ResNet-18 architectures under standard training settings.

Table 12: Dataset specifications and the respective test accuracies on ResNet-18.

| Datasets | #Classes | Training / Test Size | Image Dimensions | Clean Accuracy (%) |
|---|---|---|---|---|
| CIFAR-10 (Krizhevsky et al., 2009) | 10 | 50,000 / 10,000 | $32\times32\times3$ | 94.32 |
| CIFAR-100 (Krizhevsky et al., 2009) | 100 | 50,000 / 10,000 | $32\times32\times3$ | 75.36 |
| SVHN (Netzer et al., 2011) | 10 | 73,257 / 26,032 | $32\times32\times3$ | 96.03 |
| ImageNet-subset (Deng et al., 2009) | 100 | 20,000 / 4,000 | $224\times224\times3$ | 64.18 |

Table 13 presents the default hyperparameters for all availability poisoning attacks implemented in APBench.

Table 13: Default hyperparameter settings of attack methods.

| Methods | Hyperparameter | Settings |
|---|---|---|
| DC | Perturbation | $\ell_\infty = 8/255$ |
| | Pre-trained model | Official pretrained |
| NTGA | Perturbation | $\ell_\infty = 8/255$ |
| | Poisoned dataset | Official pretrained CIFAR-10 CNN (best) |
| EM | Perturbation | $\ell_\infty = 8/255$ |
| | Perturbation type | Sample-wise |
| | Stopping error rate | 0.01 |
| | Learning rate | 0.1 |
| | Batch size | 128 |
| | Optimizer | SGD |
| HYPO | Perturbation | $\ell_\infty = 8/255$ |
| | Step size | $\ell_\infty = 0.8/255$ |
| TAP | Perturbation | $\ell_\infty = 8/255$ |
| REM | Perturbation | $\ell_\infty = 8/255$ |
| | Perturbation type | Sample-wise |
| | Stopping error rate | 0.01 |
| | Learning rate | 0.1 |
| | Batch size | 128 |
| | Optimizer | SGD |
| | Adversarial training perturbation | $\ell_\infty = 4/255$ |
| LSP | Perturbation | $\ell_2 = 1.30$ (Project from $\ell_\infty = 6/255$) |
| | Patch size | 8 for CIFAR-10/100 and SVHN; 32 for ImageNet |
| AR | Perturbation | $\ell_2 = 1.00$ |
| | Default hyperparameters | GitHub: psandovalsegura/autoregressive-poisoning |
| OPS | Perturbation | $\ell_0 = 1$ |
| | Perturbation type | Sample-wise |
| | Default hyperparameters | GitHub: cychomatica/One-Pixel-Shotcut |
| UCL | Perturbation | $\ell_\infty = 8/255$ |
| | Poisoned dataset | GitHub: kaiwenzha/contrastive-poisoning |
| TUE | Perturbation | $\ell_\infty = 8/255$ |
| | Poisoned dataset | GitHub: renjie3/TUE |

## B  Additional Results

**Results with statistical errors** Tables 16 and 17 extends upon Table 3 by showing the detailed test accuracies of models trained on poisoned CIFAR-10 datasets, including statistical errors of 3 independent runs for each experiment.

Table 14: Default training hyperparameter settings.

| Datasets | Hyperparameter | Settings |
|---|---|---|
| CIFAR-10/-100 | Optimizer | SGD |
| | Momentum | 0.9 |
| | Weight-decay | 0.0005 |
| | Batch size | 128 |
| | Standard Augmentations | Random crop, random horizontal flip |
| | Initial learning rate | 0.1 |
| SVHN | Optimizer | SGD |
| | Momentum | 0.9 |
| | Weight-decay | 0.0005 |
| | Batch size | 128 |
| | Standard augmentations | None |
| | Initial learning rate | 0.1 |
| ImageNet-100 | Optimizer | SGD |
| | Momentum | 0.9 |
| | Weight-decay | 0.0005 |
| | Batch size | 256 |
| | Standard augmentations | Random crop, horizontal flip, and color jitter |
| | Initial learning rate | 0.1 |

Table 15: Default hyperparameter settings of defenses.

| Methods | Hyperparameter | Settings |
|---|---|---|
| Standard | Training epochs | CIFAR-10/100: 50
SVHN: 40
ImageNet-1000: 100 |
| Adversarial training (Madry et al., 2017) | Perturbation | $\ell_\infty = 8/255$ |
| | Steps size | $\ell_\infty = 2/255$ |
| | PGD steps | 10 |
| | Training epochs | 200 |
| CutOut (DeVries & Taylor, 2017)
CutMix (Yun et al., 2019)
MixUp (Zhang et al., 2018) | Training epochs | 200 |
| Gaussian (Liu et al., 2023) | Kernel size | 3 |
| | Standard deviation | 0.1 |
| | Training epochs | 200 |
| JPEG (Liu et al., 2023) | Quality | 10 |
| | Training epochs | 200 |
| BDR (Liu et al., 2023) | Number of bits | 2 |
| | Training epochs | 200 |
| UEraser-Lite (Qin et al., 2023) | PlasmaBrightness / PlasmaContrast | p = 0.5 |
| | ChannelShuffle | p = 0.5 |
| | Training epochs | 200 |
| UEraser-Max (Qin et al., 2023) | PlasmaBrightness / PlasmaContrast | p = 0.5 |
| | ChannelShuffle | p = 0.5 |
| | Number of Repeats $K$ | 5 |
| | Training epochs | 300 |
| AVATAR (Dolatabadi et al., 2023) | Diffusion sampler | Score-SDE |
| | Starting step / Total diffusion steps | 60 / 1000 |
| | Pre-trained model | GitHub: yang-song/score_sde_pytorch |
| | Training epochs | 200 |

**Partial poisoning** In addition to the discussion on partial poisoning in Section 4, we provide the results in terms of accuracies under defenses in Figure 5.

Table 16: Detailed test accuracies (%) of models trained on poisoned CIFAR-10 datasets. This table is an extension of Table 3 and further includes an error range of 3 separate runs for each experiment. The model trained on a clean CIFAR-10 dataset attains an accuracy of 94.32%. "ST" – Standard training, "CO" – CutOut, "CM" – CutMix, "MU" – MixUp, and "Gauss." – Gaussian.

| Method | ST | CO | CM | MU | Gauss. |
|---|---|---|---|---|---|
| DC | $15.19_{\pm1.87}$ | $19.94_{\pm1.90}$ | $17.91_{\pm2.66}$ | $25.07_{\pm2.74}$ | $16.10_{\pm1.07}$ |
| EM | $20.78_{\pm1.03}$ | $18.79_{\pm2.66}$ | $22.28_{\pm2.47}$ | $31.14_{\pm3.51}$ | $14.71_{\pm0.66}$ |
| REM | $17.47_{\pm2.04}$ | $21.96_{\pm3.72}$ | $26.22_{\pm2.86}$ | $43.07_{\pm4.36}$ | $21.80_{\pm0.94}$ |
| HYPO | $70.38_{\pm1.79}$ | $69.04_{\pm1.33}$ | $67.12_{\pm3.27}$ | $74.25_{\pm2.60}$ | $62.17_{\pm1.03}$ |
| NTGA | $22.76_{\pm0.67}$ | $13.78_{\pm0.75}$ | $12.91_{\pm1.22}$ | $20.59_{\pm2.41}$ | $19.95_{\pm1.26}$ |
| TAP | $6.27_{\pm0.48}$ | $9.88_{\pm0.71}$ | $14.21_{\pm2.14}$ | $15.46_{\pm1.77}$ | $7.88_{\pm0.66}$ |
| LSP | $13.06_{\pm0.74}$ | $14.96_{\pm1.02}$ | $17.69_{\pm1.37}$ | $18.77_{\pm2.12}$ | $18.61_{\pm0.82}$ |
| AR | $11.74_{\pm0.37}$ | $10.95_{\pm0.70}$ | $12.60_{\pm1.02}$ | $14.15_{\pm1.28}$ | $13.83_{\pm1.70}$ |
| OPS | $14.69_{\pm0.55}$ | $52.98_{\pm2.49}$ | $64.72_{\pm2.70}$ | $49.27_{\pm2.66}$ | $13.38_{\pm0.31}$ |

Table 17: Table 16 continued. "BDR" – Bit-Depth Reduction, "GS" – Grayscale, "JPEG" – JPEG Compression, "U-Max" – UEraser-Max, "AVA" – AVATAR, and "AT" – adversarial training.

| Method | BDR | GS | JPEG | U-Max | AVA | AT |
|---|---|---|---|---|---|---|
| DC | $67.73_{\pm2.77}$ | $85.55_{\pm2.04}$ | $83.57_{\pm2.61}$ | $92.17_{\pm0.97}$ | $82.10_{\pm2.20}$ | $76.85_{\pm1.79}$ |
| EM | $37.94_{\pm2.64}$ | $92.03_{\pm0.37}$ | $80.72_{\pm1.22}$ | $93.61_{\pm0.48}$ | $75.62_{\pm2.16}$ | $82.51_{\pm1.24}$ |
| REM | $58.60_{\pm2.35}$ | $92.27_{\pm0.29}$ | $85.44_{\pm2.10}$ | $92.43_{\pm0.61}$ | $82.42_{\pm1.47}$ | $77.46_{\pm1.66}$ |
| HYPO | $74.82_{\pm2.18}$ | $63.35_{\pm0.88}$ | $85.21_{\pm1.35}$ | $88.44_{\pm0.71}$ | $85.94_{\pm2.06}$ | $81.49_{\pm2.37}$ |
| NTGA | $59.32_{\pm1.96}$ | $70.41_{\pm2.67}$ | $68.72_{\pm3.37}$ | $86.78_{\pm1.64}$ | $86.22_{\pm2.07}$ | $69.70_{\pm2.66}$ |
| TAP | $70.75_{\pm1.46}$ | $11.01_{\pm0.67}$ | $84.08_{\pm2.36}$ | $79.05_{\pm1.04}$ | $87.75_{\pm2.52}$ | $79.92_{\pm1.87}$ |
| LSP | $53.86_{\pm2.40}$ | $64.70_{\pm3.29}$ | $80.14_{\pm2.51}$ | $92.83_{\pm1.27}$ | $76.90_{\pm1.09}$ | $81.38_{\pm1.92}$ |
| AR | $36.14_{\pm2.04}$ | $35.17_{\pm1.77}$ | $84.75_{\pm2.27}$ | $90.12_{\pm1.89}$ | $88.60_{\pm2.33}$ | $81.15_{\pm1.94}$ |
| OPS | $37.32_{\pm1.87}$ | $19.88_{\pm0.79}$ | $78.48_{\pm0.94}$ | $77.99_{\pm1.30}$ | $66.16_{\pm2.13}$ | $14.95_{\pm0.67}$ |

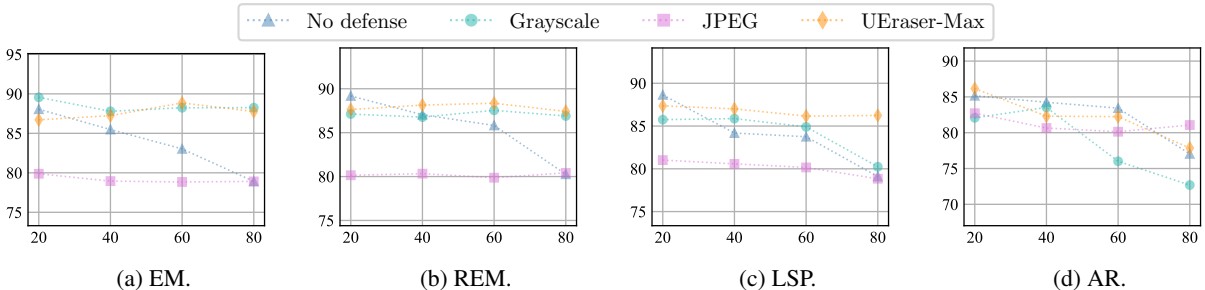

Figure 5: The efficacy in test accuracies (%, vertical axes) of defenses (No defense, Grayscale, JPEG, and UEraser-Max) against different partial poisoning attacks including EM (a), REM (b), LSP (c), and AR (d) with poisoning ratios (horizontal axes) ranging from 20% to 80%.

## B.1 Visualizations

**Grad-CAM and Shapley visualizations** Gradient-weighted class activation mapping (Grad-CAM) (Selvaraju et al., 2017) and Shapley value map (Lundberg & Lee, 2017) are commonly used image analysis tools that visualize the contributions of different pixels in an image to the model's predictions. From the Shapley value map and Grad-CAM visualizations (Figure 6), we can observe discernible changes in activation features in the poisoned model relative

to the clean model. AVATAR showed activation features most similar to the clean model compared to other defense mechanisms, as it aims to restore the original features while disrupting the APA perturbations. In contrast, models applying the other defense strategies typically have different activation features than clean models. This discrepancy implies that image preprocessing or augmentations may modify the inherent feature extraction from the original images instead of restoring them.

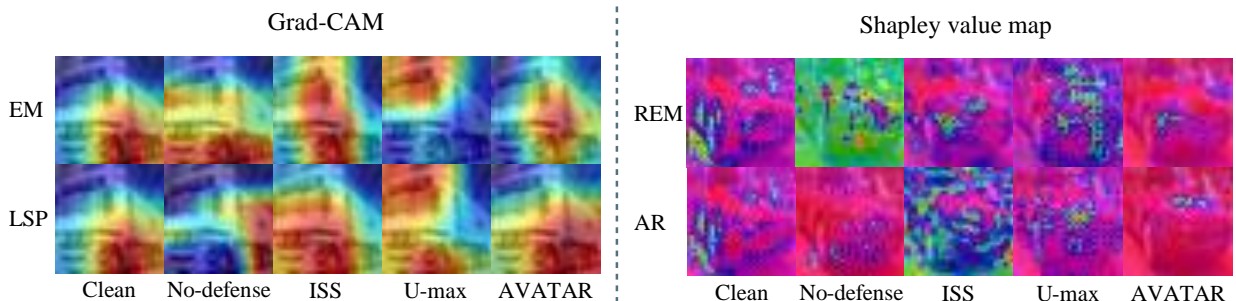

Figure 6: Grad-CAM and Shapley value map visualizations of regions contributed to model decision under different attack methods and defense methods with ResNet-18. (Left) Grad-CAM visualizations of EM and LSP attacks. (Right) Shapley value map visualizations of REM and AR attacks.

**T-SNE visualizations** Figure 8 shows the t-SNE visualization Figure 7 of the models' feature representations on the clean test set for CIFAR-10. Notably, models without defenses struggle to create coherent class clusters, although there exist spatial variations in class frequency. In contrast, models equipped with defenses display a feature distribution akin to the clean baseline. Models with higher clean accuracies also often exhibit better separated clusters.

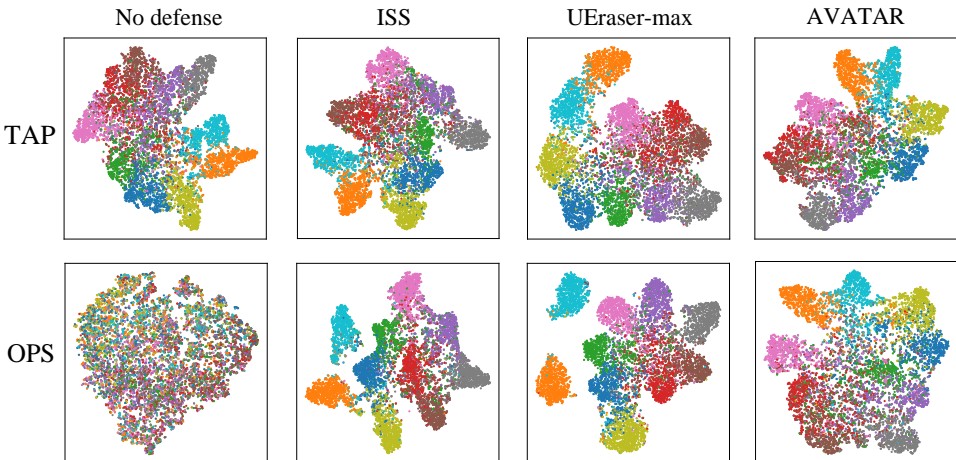

Figure 7: The t-SNE visualization of the models' feature representations on the clean test set. Note that without defenses, the feature representations of the poisoned models are mostly scrambled as the models struggle to learn useful features.

## C   Licenses

APBench is open source, and the source code will be made available upon publication. Table 18 provides the licenses of the derived implementations of the original algorithms and datasets.

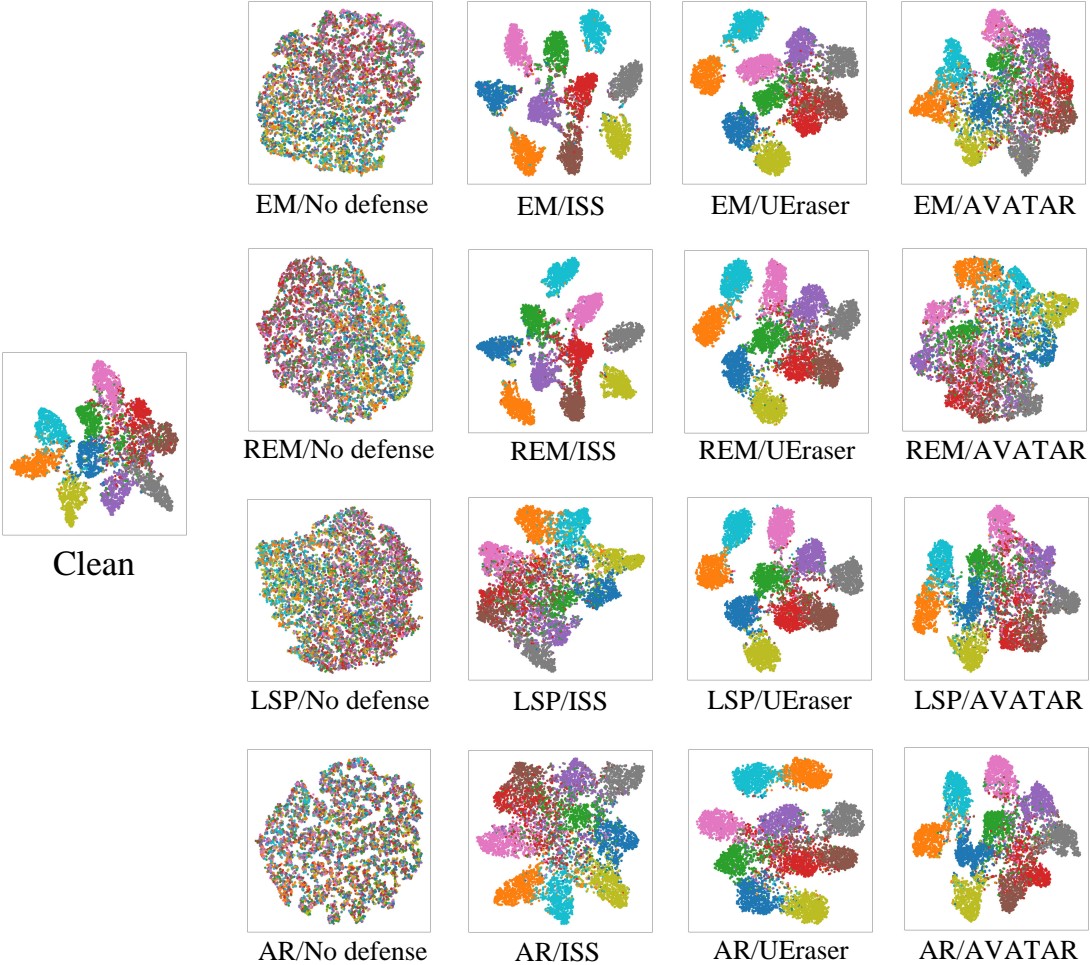

Figure 8: The t-SNE visualization of the models' feature representations on the clean test set under additional attacks for CIFAR-10. "UEraser" denotes "UEraser-Max".

Table 18: Licenses of the datasets and codebases used in this paper.

| Name | License | URL |
|---|---|---|
| PyTorch | BSD | GitHub: pytorch/pytorch |
| **Attacks** | | |
| DC | — | GitHub: kingfengji/DeepConfuse |
| NTGA | Apache-2.0 | GitHub: lionelmessi6410/ntga |
| EM | MIT | GitHub: HanxunH/Unlearnable-Examples |
| HYPO | MIT | GitHub: TLMichael/Delusive-Adversary |
| TAP | MIT | GitHub: lhfowl/adversarial_poisons |
| REM | MIT | GitHub: fshp971/robust-unlearnable-examples |
| LSP | — | GitHub: dayu11/Availability-Attacks-Create-Shortcuts |
| AR | MIT | GitHub: psandovalsegura/autoregressive-poisoning |
| OPS | Apache-2.0 | GitHub: cychomatica/One-Pixel-Shotcut |
| UCL | MIT | GitHub: kaiwenzha/contrastive-poisoning |
| TUE | — | GitHub: renjie3/TUE |
| **Defenses** | | |
| ISS | — | GitHub: liuzrcc/ImageShortcutSqueezing |
| DiffPure | NVIDIA | GitHub: NVlabs/DiffPure |
| **Datasets** | | |
| CIFAR-10 | — | https://www.cs.toronto.edu/~kriz/cifar.html |
| CIFAR-100 | — | https://www.cs.toronto.edu/~kriz/cifar.html |
| SVHN | — | http://ufldl.stanford.edu/housenumbers |
| ImageNet-100 | — | GitHub: TerryLoveMl/ImageNet-100-datasets |

