# OpenReview forum: "APBench: A Unified Availability Poisoning Attack and Defenses Benchmark"
_TMLR — Accepted by TMLR_

### Review · Reviewer_uKf3 · 2024-05-28

**Summary Of Contributions:**

This paper is mainly about the authors’ development of a benchmark that evaluates the effectiveness of availability poisoning. Specifically, the authors focus on the availability poisoning attack (APA) that is used to improve data privacy against unauthorized training attempts and the defense against APA that tries to circumvent such privacy protection.

The authors formally describe the availability poisoning attack and the defense against it, with examples of existing attacks and defenses. They also point out the lack of benchmarks for availability poisoning. Then, they introduce the APBench benchmark, which can assess the performance of APA against existing defenses.

Then, the authors present a set of experimental results evaluated from the APBench benchmark. They explore both standard scenarios (commonly assumed by the researchers) and challenging scenarios (that modify the common scenarios to reflect more realistic variations). From these experimental explorations, the authors demonstrate three non-trivial findings about adversarial poisoning attacks and defenses.

**Audience:**

Yes

**Broader Impact Concerns:**

I don’t see a particular concern about the broader impact of this paper.

**Claims And Evidence:**

Yes

**Requested Changes:**

### Evaluations section
1. The Evaluations section should explain why the authors want to run experiments. I suggest writing a few paragraphs describing the experiments' goals. It is also better to present the **Key Takeaways** at the beginning of this section.
2. Because the experiments involve many attack-defense combinations, the tables are difficult to interpret. To facilitate interpretation, highlight some key values (that demonstrate the authors' desired results) in boldface letters.
3. In Tables 3 and 4, it is better to match the attack orderings to those of Table 1 and the defense orderings to those of Table 2. This makes it easier to look up the original papers and follow chronological order so readers can check whether or not recent attacks/defenses perform better. Also, if the vertical line in Tables 3 and 4 (between columns “MixUp” and “Gaussian”) is intended to follow the separation in Table 2, there should be another vertical line separating the Data preprocessing category and the Training-phase defense category.
4. IIUC, the authors regard “Standard” data augmentation as one of the defense methods in Table 2. If so, the authors should present evaluation results for “No defense” (that does not do any augmentation/preprocessing) as the baseline in Tables 3 and 4.

### Minor changes
1. Overall, the paper writing is good, but there are small mistakes. I suggest the authors proofread the paper once again.
   * In Section 2.3, the last sentence of the second paragraph should use in-text citations. For details about the citation styles, please read the [TMLR LaTeX stylefile and template](https://github.com/JmlrOrg/tmlr-style-file/archive/refs/heads/main.zip) in [the guidelines for authors](https://www.jmlr.org/tmlr/author-guide.html).
   * On Page 6, “... is not usually considered a good one. mainly due to the fact…” should not have a period in the middle of the sentence.
2. While I don’t think that there is a clear standard for writing the **Reproducibility Statement** and the **Ethics Statement**, they look too short to be separate Sections. One possible suggestion is to write a dedicated section for miscellaneous **Discussions**, containing subsections for **Future research direction** (in the last paragraph of Section 4), **Limitations** (Section D), **Reproducibility Statement** (Section 6), and **Ethics Statement** (Section 7).

**Strengths And Weaknesses:**

### Strength
1. The paper is well-motivated, and the benchmark covers many existing attacks and defenses.
2. The authors present an extensive amount of evaluation, and the evaluation demonstrates the authors’ own findings.
3. The authors explore different challenge scenarios that can happen in real-world cases.

### Weaknesses
1. The writing of the paper can be improved further, making it more readable and organized. Please see **Requested Changes** for details.

---

> ### Author Response · Authors · 2024-06-26
> **Thank you for reviewing our paper and we would like to address your concerns below.**
>
> > It is also better to present the Key Takeaways
> > at the beginning of this section.
>
> Thank you for your constructive suggestions!
> We have added an evaluation objective summary in Section 4.1,
> and moved the key takeaways to the beginning
> of the evaluation section in the updated version.
>
> > To facilitate interpretation,
> > highlight some key values in boldface letters.
>
> We have highlighted best defenses
> in each category in bold (highest accuracy for the same attack),
> and best attacks with underline (lowest accuracy for the same defense)
> in Tables 3-10.
>
> > Table changes: In Tables 3 and 4, it is better
> > to match the attack orderings ...
> > ... Table 2, there should be another vertical line ...
>
> Thank you for your detailed suggestion.
> We have incorporated this modification
> in the updated version.
>
> >  Present evaluation results for "No defense" in Tables 3-4.
>
> Thank you for pointing this out to us!
> We have included the results of "No Defense" in the updated version.
>
> > Minor changes.
>
> Thanks again for your detailed suggestions.
> We have made thorough proofreading,
> and corrected numerous typos in the updated version.
> In addition,
> some texts are expanded for clarity,
> as shown with color highlights.
> We have also merged the sections
> (Future research directions, limitations,
> reproducibility statement, and ethics statement)
> into Section 5 "Discussion" in the updated version.

---

### Review · Reviewer_ZpQS · 2024-06-03

**Summary Of Contributions:**

This paper develops a benchmark--APBench for assessing the efficacy of adversarial poisoning, including Attacks and Defenses. APBench consists of 9 availability poisoning attacks, 8 defense algorithms, and 4 conventional data augmentation techniques. It conducts experiments with varying different poisoning ratios, and evaluates the attacks on multiple datasets and their transferability across model architectures. The results reveals the glaring inadequacy of existing attacks in safeguarding individual privacy.

**Audience:**

Yes

**Claims And Evidence:**

No

**Requested Changes:**

1.	Provide more evidence to support the claims or reword the claims.
2.	Reword the related work with better presentation.

**Strengths And Weaknesses:**

**Strengths:**

The topic is interesting and this paper provide a Benchmark, which might be useful for the researchers.
This paper also conducts many experiments, some of which are valuable.


**Weaknesses:**

1.This paper overclaims the contributions. This paper uses strong terms to describe the proposed benchmark, e.g., “this comprehensive suite allows us to evaluate the effectiveness of the poisoning attacks thoroughly”, “with a clear understanding of the current progress…”, “An open source benchmark for state-of-the-art availability poisoning attacks and defencses.”. However, I cannot find enough evidence to support these terms:
(1)	Why this paper uses these availability poisoning attacks and defenses methods? Why they are comprehensive for evaluation? By the way, how does the implementation (or the results) of the methods related to the original implementation (results) of the corresponding papers? Better or worse? Or not comparable? I believe this paper should provide these information to support its claims.
(2)	This paper claims “the experiments are conducted on 5-mainstream models , ResNet-18, ResNet-50, MobileNetV2 and DenseNet-121”, why not conduct experiments on ViTs here? Apparently, Vits have significant differences compared to these traditional CNNs. Besides, why this paper claims in Section 1“also extensively examined scenarios of partial poisoning, increased perturbations, the transferability of attacks to 4 CNN and 2 ViT models under various defenses, and unsupervised learning…”?
(3)	If this paper claims “with a clear understanding of the current progress…”, I would understand this paper will describe the related work like a survey, but it not.

2.This paper should re-organize the description of the related work in page 2 (the last paragraph). In the current description, this paper only describes each method, but I believe this paper should well organize them with better logic.


3.This paper should provide more clarification in its description. For example:
(1) in Section 2.1, this paper states that “The attacker’s goal is to thus render their data unlearnable with perturbation”, What is “their data unlearnable”? How to define “unlearnable”? Does this definition depends on tasks?
(2) In section 4, it states “To ensure a fair comparison between attack and defense methods, we used only the basic version of training for each model”, what does the meaning of “he basic version of training for each model”? why this can ensure “a fair comparison between attack and defense methods”. Please clarify it.

---

> ### Author Response · Authors · 2024-06-26
> **Thank you for reviewing our paper and we would like to address your concerns below.**
>
> > 1. This paper overclaims the contributions.
>
> We have tuned down the claims in the revised version
> to better reflect the contributions of our work.
> We have also revised the abstract and introduction
> to better align with the contributions of our work.
>
> > 1.1: Why this paper
> > uses these availability poisoning attacks and defenses methods?
> > Why they are comprehensive for evaluation?
>
> This paper
> chooses to evaluate availability poisoning attacks and defenses
> as they represent an emerging research direction
> that focuses on the privacy protection
> against unsolicited model training.
> This research direction has not been systematically evaluated
> in related benchmarks,
> and most existing works
> do not have a standardized threat models or evaluation protocols.
> We aim to fill this gap with APBench.
> The availability poisoning attack and defense baselines
> that we have selected from recent literature
> are mostly proposed in the past two years,
> and are widely regarded
> as the current state-of-the-art approaches.
> If you have any suggestions
> for other methods to include in our benchmark,
> we would be happy to consider them.
>
> > 1.1 (continued):
> > How does the implementation (or the results) of the methods
> > related to the original implementation (results) of the corresponding papers?
> > Better or worse?
> > Or not comparable?
>
> We have derived the implementations of the methods
> based on their original sources,
> and have made efforts to align our results
> with the original results as closely as possible.
> As the original papers
> may not have consistent evaluation protocols,
> we have made necessary, but minimal, adjustments
> to ensure a fair comparison among all methods.
>
> > 1.2: Why no ViTs?
> > Besides, why this paper claims in Section 1“also extensively examined scenarios of partial poisoning, increased perturbations, the transferability of attacks to 4 CNN and 2 ViT models under various defenses, and unsupervised learning…”?
>
> Thank you for your suggestion.
> We kindly note that perhaps you may have missed the ViT results,
> which are included in the last columns of Table 6,
> and are discussed in Section 4.3 --
> "Attack transferrability across models".
>
> For clarity,
> please refer to Section 4.3 for the results
> on partial poisoning, increased perturbations,
> transferability, and unsupervised learning.
>
> > 1.3 and 2: Re-organize the description of the related work
>
> Thank you for your constructive suggestions.
> Since we are page-limited,
> in the updated version,
> we have provided an extended overview
> of the relevant attack and defense methods,
> which can be found in Sections 2.1 and 2.2, respectively.
>
> > What is "their data unlearnable"?
> > How to define "unlearnable"?
> > Does this definition depends on tasks?
>
> The word "unlearnable" (used as an adjective here),
> in "render their data unlearnable",
> refers to the (extent of) inability of the model
> to learn the original features of the data.
>
> The definition is typically task-agnostic,
> but the surrogate-based algorithms
> typically employ the same loss function as the original model.
> "Unlearnability" can thus be measured
> by the model's original performance metric
> after training on the poisoned data.
>
> > What does the meaning of "the basic version of training for each model"?
> > why this can ensure "a fair comparison between attack and defense methods"
>
> By the "basic version",
> we mean that the models considered
> in our evaluation are the publicly-available versions
> with wide adoption in the community.
> As we do not make any modifications to the model implementations
> and use widely-used models,
> it helps to ensure the generality and fairness of our evaluation.
> We have also updated the text for clarity.

---

> > ### Comment · Reviewer_ZpQS · 2024-07-16
> > **In terms of the Vits for Standard Scenario**
> >
> > Clarfication:
> > In terms of the ViT comments (This paper claims “the experiments are conducted on 5-mainstream models , ResNet-18, ResNet-50, MobileNetV2 and DenseNet-121”, why not conduct experiments on ViTs here?"),
> >
> > I mean why not conduct experiments using Vits on standard scenario. (I noted this paper uses Vits for "Attack transferability across models")

---

> > > ### Author Response · Authors · 2024-07-16
> > > **Response to the suggestion of ViTs in the standard scenario**
> > >
> > > Thank you for your suggestion! We kindly note that the standard scenario protocol considers models that are trained from scratch, and ViTs typically do not achieve satisfactory accuracies on the clean CIFAR-10 training set (~80% test accuracy) when trained from scratch. We thus did not carry out the experiments, since this protocol is not representative of how ViTs are trained. Nevertheless, we are running additional experiments to extend the standard scenario evaluation to ViT models.

---

> ### Author Response · Authors · 2024-07-19
> **Update on additional experiments on ViTs**
>
> We have conducted experiments using ViT-small models under the standard scenario (i.e., being both surrogate and target), and present the results below, now included the paper (the newly added Table 4). For reference, the model trained from scratch on a clean CIFAR-10 dataset attains an accuracy of 84.66%.
>
> |      |   ND  |   ST  |   CO  |   CM  |   MU  | Gauss |  BDR  |   GS  | JPEG |  AVA  |   ES  | U-Max |   AT  |
> |------|-------|-------|-------|-------|-------|-------|-------|-------|------|-------|-------|-------|-------|
> | DC   | 20.37 | 22.94 | 26.03 | 32.80 | 33.64 | 26.06 | 55.47 | 72.45 |71.82 | 79.16 | 26.49 | 82.89 | 77.26 |
> | EM   | 36.82 | 37.14 | 38.79 | 40.30 | 46.91 | 36.29 | 54.59 | 59.38 |70.69 | 72.81 | 41.29 | 81.64 | 78.03 |
> | REM  | 30.61 | 31.72 | 29.90 | 34.68 | 44.99 | 33.14 | 50.59 | 80.29 |74.63 | 75.52 | 42.88 | 82.50 | 79.32 |
> | HYPO | 74.47 | 75.23 | 75.76 | 74.60 | 76.16 | 75.12 | 79.21 | 63.62 |75.77 | 83.15 | 67.29 | 83.50 | 78.62 |
> | TAP | 22.38 | 22.93 | 22.65 | 24.86 | 27.82 | 23.47 | 40.03 | 37.75 |67.26 | 73.26 | 31.17 | 69.83 | 76.55 |
> | NTGA | 26.40 | 28.27 | 22.14 | 23.39 | 39.58 | 25.57 | 52.38 | 56.92 |53.94 | 74.88 | 33.26 | 76.39 | 77.08 |
> | LSP  | 27.14 | 29.06 | 30.17 | 32.66 | 28.38 | 33.07 | 41.17 | 59.34 |68.07 | 66.74 | 32.69 | 87.01 | 76.94 |
> | AR   | 22.37 | 25.04 | 26.92 | 21.18 | 30.67 | 25.48 | 37.04 | 38.90 |74.77 | 78.64 | 45.54 | 63.90 | 75.62 |
> | OPS  | 18.16 | 20.84 | 61.27 | 76.59 | 34.58 | 32.39 | 45.71 | 31.60 |69.31 | 59.26 | 22.63 | 66.72 | 24.30 |

---

> > ### Comment · Reviewer_ZpQS · 2024-07-24
> > **Acknowledge the response**
> >
> > Thanks for the additional experiments, and my concerns are this point is relieved.

---

### Review · Reviewer_2T9D · 2024-06-11

**Summary Of Contributions:**

This paper discusses benchmarks for availability poisoning attacks, which involve injecting perturbations into training samples to prevent unauthorized trainers from using these data to train models. Specifically, this paper presents APBench, an open-source benchmark that encompasses a range of poisoning attacks and defense strategies. The experiments reveal limitations in the current attack methods' effectiveness at safeguarding privacy. APBench is designed to foster equitable and reproducible evaluation, propel the advance of privacy-preserving availability poisoning attacks and defense mechanisms.

**Audience:**

Yes

**Broader Impact Concerns:**

No broader impact concerns.

**Claims And Evidence:**

Yes

**Requested Changes:**

See weakness above.

**Strengths And Weaknesses:**

Strengths:
- Standardization and Reproducibility: APBench standardizes the experimental setup, ensuring fair and reproducible evaluations. This addresses the issue of varying experimental setups in previous research, providing a consistent platform for comparison.
- Wide Range of Attacks and Defenses: The benchmark covers 9 availability poisoning attacks, 8 defense algorithms, and 4 conventional data augmentation techniques. This extensive coverage provides a broad perspective on the effectiveness of different methods.
- Focus on Privacy: The research highlights the inadequacy of existing attacks in safeguarding individual privacy, emphasizing the importance of developing more effective methods. This focus on privacy protection aligns with the growing concerns regarding data security.

Weaknesses:
- Limitations of datasets: The datasets used in the paper are mainly traditional natural image datasets, such as CIFAR-10, CIFAR-100, SVHN, and a subset of ImageNet. Although these datasets are widely used in visual tasks, their diversity and complexity may not be sufficient to comprehensively evaluate various data poisoning attacks and defense methods. Increasing the use of mixed (natural + generated images) datasets may provide a more comprehensive evaluation.
- It appears that the APA's effectiveness in privacy protection is gauged by the extent of reduction in model performance, akin to untargeted poisoning attacks that aim to degrade model performance by causing specific samples to be misclassified. Perhaps this paper can be compared with the relevant benchmark [1].

[1]Just How Toxic is Data Poisoning? A Unified Benchmark for Backdoor and Data Poisoning Attacks abstract

---

> ### Author Response · Authors · 2024-06-26
> **Thank you for reviewing our paper and we would like to address your concerns below.**
>
> > Limitations of datasets
>
> We agree that mixed datasets
> can potentially provide a broader scope for evaluation.
> Since the primary focus of our submission
> pertains user privacy,
> this benchmark was designed to evaluate the effectiveness
> of availability poisoning and defense mechanisms
> on user data,
> which are in general natural ---
> rather than synthetic --- images.
> If you feel that mixed dataset
> is essential to provide a more comprehensive evaluation,
> we kindly request your guidance
> in identifying a suitable natural + generated images dataset.
> We will conduct additional experiments accordingly.
>
> > Comparison to "Just How Toxic is Data Poisoning"
>
> Thank you for your constructive suggestion.
> Since this benchmark was not introduced
> in the original submission,
> we have included it in the revised manuscript
> in Section 2.3 Related Benchmarks.
> Notably,
> this benchmark focuses on the effectiveness
> of traditional data poisoning attacks
> in the context of real-world challenges.
> Interestingly,
> it examines the *triggerless* attack scenario,
> where it forces the model
> to misclassify a *specific* target image,
> opening a backdoor in the model
> for post-training attack opportunities.
> In contrast,
> attacks in APBench
> aims to prevent the model from learning
> from features present
> in the *entire* poisoned set of images
> to preserve user privacy.

---

### Author Response · Authors · 2024-06-26
**Manuscript has been updated with colors highlighting changes**

We sincerely thank the reviewers
for their positive feedback and constructive comments.
We appreciate your recognition
of its standardization and reproducibility (2T9D),
extensive (all reviewers) and valuable (ZpQS) evaluations,
well-motivated (uKf3) and interesting (ZpQS)
nature of the problem,
and overall writing clarity (uKf3).
We are grateful for your suggestions
and continued to improve the paper
based on your feedback.

We highlight the changes made in the revised version
requested by each reviewer
with different colors in the manuscript,
and attribute them to the respective reviewers
by your assigned IDs.
In addition,
we have made general improvements to the paper
which are highlighted in green.

---

### Decision · Action_Editor_5EUE · 2024-07-31

**Recommendation:** Accept as is

**Comment:**

All three reviewers unanimously think the paper should be accepted after the modifications made by the authors (2 x Accept, 1 x Leaning Accept). Also, there has been a lengthy discussion with Reviewer ZpQS on the experiments.

**Audience:**

Yes, as pointed out by the reviewers, this (availabilty poisoning attacks and defences) is a growing field within ML and the benchmarks will potentially be used by many readers of TMLR.

**Claims And Evidence:**

The paper builds a benchmark, “APBench” for assessing the efficacy of availability poisoning attacks where the dataset is aimed to be made useless for training with small perturbations. The benchmark consists of 9 state-of-the-art availability poisoning attacks, 8 defense algorithms, and 4 conventional data augmentation techniques. Experiments using the defence methods also reveal that they are not able to protect the privacy of the data, which is an independent observation and contribution of the paper. The field (availabilty poisoning attacks and defences) is a growing field and the aim of the paper is to provide benchmarks for experimental evaluations.